# Stakeholders' Experiences and Perceptions of the Provision and Practice of Language Support for Ethnic Minority School Children in Japan

## Michi Saki

Department of English, Faculty of Culture and Representation, Doshisha Women's College of Liberal Arts, Kyoto 610-0395, Japan; msaki@dwc.doshisha.ac.jp

**Abstract:** This study examines experiences and perceptions concerning the provision of Japanese language support for ethnic minority school children between the ages of 6 and 12 enrolled in public elementary schools in a city located in the Kansai region of Japan (hereinafter referred to as "City M"). This paper will focus in particular on interpreting the experiences and perspectives of language support teachers, volunteer interpreters, mother-tongue language supporters (hereafter referred to as MTLS) as well as three principals of three public elementary schools located in particularly different areas of the City M. Each of the school's history, backgrounds and current situations are varied and unique. One-on-one interviews of 40 to 60 min in length were conducted with a total of 9 participants consisting of public elementary school principals and Japanese language support teachers. These results of the data collection provided a deeper understanding and explanation of the reasons behind current trends and challenges regarding the accessibility, implementation, provision, and practice of language learning support for ethnic minority school children. The findings from this research will increase awareness of current issues faced by practitioners supporting ethnic minority children in their learning. The research findings also provide insight into what obstacles need to be overcome in order to provide adequate, sufficient, and sustainable educational support for children from different cultural and linguistic backgrounds within the mainstream Japanese education system.

**Keywords:** ethnic minority children; Japanese language support; provision; practice; deficiency; disconnect; disparity; sustainability

## 1. Introduction

### 1.1. A Growing Diversity

Japan has been one of the few advanced industrialized nations in the world not to openly invite migrant workers in large numbers into the country, with one reason being to preserve ethnic homogeneity and social harmony. Over the past two decades, however, the country's declining population and shortages of unskilled labour have led Japan to open its doors to migrant workers as a measure aimed at addressing the country's labour shortage. The Japanese government has been revising its immigration laws since the 1990s and has been loosening immigration restrictions to allow more foreign migrants to enter the country. The rising number of foreign nationals in Japan has brought an increase in ethnic diversity to Japanese society, and as a result, a steadily growing number of ethnically diverse children have enrolled in Japanese public schools. Transnational tendencies, international marriage, and overall globalization of the country, resulting in the creation of a more diverse ethnic population, create many challenges for multicultural families in Japan. One of the biggest challenges is the education of their children [1–5]. In the past 25 years, research on ethnic minority children and Japanese language and learning support has been conducted in the Kansai area, especially in prefectures where there are denser populations [6–9]. According to recent statistics, the number of ethnic minority children in City M, however, seems to fluctuate and be relatively small in comparison with other Kansai cities.

*1.2. Rationale and Background of Research Framework*

At the start of this research project, I planned to narrowly focus on ethnic minority school children enrolled in public elementary schools in City M, and to examine their learning issues and overall social well-being at school. However, when analyzing the data collected, I found that the concepts and categories that emerged from the data tend to center around issues of Japanese language support, the challenges of language support providers, and the obstacles faced by many stakeholders who are supporting ethnic minority children from various backgrounds and with different language capabilities. At this point, I realized that I could not limit my research focus to the narrow category of foreign-born, immigrant children, as they are not the only group of ethnic minority children who lack the language skills to succeed in academic subjects at school. Though the data findings contained significant content about the identity and well-being of ethnic minority children and their public schooling in general, many of the dialogues and conversations focused heavily on issues of Japanese language and learning support services and programs. As a result, I decided that I needed to change the focus of my research framework towards the Japanese language and learning support for different categories of ethnic minority children.

In changing the parameters of my study to focus on Japanese language and learning support for ethnic minority children, I felt that I needed to determine how the term "ethnic minority child" should be defined. The following two categories of ethnic minority children have come to be the focus of this study.

1. The first category is children referred to in Japanese as gaikokujin jidō seito (foreign school children) or nyūkamā no kodomo (newcomer children), commonly translated into English as "foreign children". These are children who were born in a foreign country and have migrated with their parents to Japan. Japanese is their second language, with their first language being the mother tongue of their parent(s).
2. The second category is children who are commonly referred to in Japanese as gaikoku ni roots o motsu jidō seito or gaikoku ni tsunagaru kodomo, translated into English as "school children with foreign roots". Two other terms commonly used are idō suru kodomo and kikoku tonichi jidō seito, translated as "school children crossing borders". These terms describe a specific group of children who have been born and/or predominantly raised in Japan or have lived and been raised in both a foreign country and Japan for a significant amount of time, but who identify with a language and culture other than Japanese and therefore require Japanese language support.

I realized that my initial plan of focusing on one group of ethnically diverse children was not going to accurately represent the various experiences and perspectives of all the people who shared their stories with me. I therefore decided to define "ethnic minority children" with these two categories in order to include children existing both inside and outside pre-determined labels, and to explore their unique backgrounds, cultural and linguistic identities, learning needs, and future academic challenges in Japan.

*1.3. Research Questions*

This study seeks to explore and answer the following questions:

1. What is the current situation of the implementation, provision, and practice of Japanese language learning support for ethnic minority school children enrolled at public elementary schools in City M?
2. What are the trends and challenges facing the local actors who provide formal language instruction in ensuring the implementation, provision, and practice of Japanese language learning support for ethnic minority children in City M? Some of these actors include public elementary schools, including their specialized Japanese language support teachers, homeroom teachers, head teachers, and principals.
3. What are the trends and challenges facing the local actors who provide informal language instruction for ethnic minority children in City M? These actors include volunteer language support teachers, mother tongue language supporters, interpreters

dispatched from international exchange organizations, NPOs, grassroots organizations, and private Japanese language support groups.

### 1.4. Purpose of the Study

The primary objective of this study is to thoroughly examine the landscape of Japanese language education and learning support offered to two distinct groups of ethnic minority children in City M. This city has been witnessing a steady rise in its international population due to an influx of migrant workers, overseas business professionals, and academics. However, the diversity within its foreign communities presents significant challenges in delivering adequate assistance to children attending schools without comprehensive language support programs.

This research project delves into the personal experiences and viewpoints of diverse stakeholders situated in City M. This includes public elementary school educators, Japanese language support volunteers, local educational organizations, and both government and non-government agencies involved in Japanese language support initiatives.

In addition to the above, this study also seeks to foster meaningful conversations among various institutions, municipal bodies, and other local entities that bear the responsibility of educating ethnic minority children. The ultimate aim is to raise awareness regarding the prevalent issues concerning the implementation and execution of language learning support programs. By doing so, the study intends to encourage proactive efforts in enhancing the training and education of both pre-service and in-service teachers within public schools. Simultaneously, it strives for a balanced and effective allocation, dissemination, and management of educational resources. These combined efforts are directed towards catering to the multifaceted educational requirements of this emerging and diverse generation of ethnic minority children.

### 1.5. A Theoretical Framework for Language Support as Educational Equity

One type of framework for language support that promotes educational equity is the Culturally Responsive and Linguistically Sustaining Education (hereafter referred to as CRSE) Framework. This framework highlights the importance of recognizing and valuing students' diverse linguistic and cultural backgrounds while providing effective language support. The Culturally Responsive and Linguistically Sustaining Education (CRSE) framework has been influenced by the work of multiple educators and scholars in the field of education [10,11].

CRSE framework is intended to help education stakeholders create student-centered learning environments that affirm cultural identities; foster positive academic outcomes; develop students' abilities to connect across lines of difference; elevate historically marginalized voices. The framework was designed to support education stakeholders in developing and implementing policies that educate all students effectively and equitably, as well as provide appropriate support and services to promote positive student outcomes [10,11].

The CRSE framework aims to create an inclusive educational environment that empowers students from diverse backgrounds to succeed academically and develop a strong sense of identity. This framework involves such principles as cultural responsiveness, linguistic sustaining practices, equity and inclusion, critical pedagogy and collaboration and community engagement [10,12].

This study will adopt the CRSE framework by promoting education equity through language support, whereby emphasizing the need to provide equal opportunities for all students, regardless of their linguistic background. This framework will be used to critically analyze the issues surrounding language support systems in place, in order to illuminate widening linguistic gaps, and examine whether or not minority children have equitable access to quality education and whether or not they can fully participate in all aspects of schooling.

*1.6. Positionality of the Researcher*

The term "positionality" refers to an individual's worldview and the position they have chosen to take in relation to a specific research task; it has to do with the researcher's relationship with participants, the nature of that involvement, how much of the study's purpose will be revealed to participants, and how ethical dilemmas will be managed [13–15]. In this study, I have chosen to take the position of an "insider". I am a long-time foreign resident of City M whose mother tongue is a language other than Japanese, I am an educator, and I am also a parent of an ethnic minority child who is bilingual and considers a language other than Japanese to be her mother tongue. I believe that taking the position of an "insider" has significantly helped me to make closer connections with the participants, allowing me to become more intimate with them and their families due to my cultural and linguistic background. I felt that I could collect richer data as an "insider" than if I was considered an "outsider". Research participants may have felt it easier to open and reveal their stories and personal perspectives. However, it could also be argued that by taking the "insider" position in this research, I may have collected and processed data with a biased view, which may have limited the way the findings were interpreted and analyzed and may have influenced the conclusions I reached.

## 2. Ethnicity and Categories of Diversity

Japan has often been described by sociologists as "homogeneously minded", projecting itself as an ethnically and linguistically homogeneous nation [16–19]. Japan has never been a mono-ethnic nation, however. There have always been ethnic minorities within Japanese society: Koreans, Chinese, Ainu, Okinawans, and burakumin (descendants of feudal outcaste populations). The list of minorities in Japan continued to grow in the 1990s with the addition of many other nationalities and ethnicities spreading throughout Japan, including Japanese descendants from South America. There has been a particular surge of cultural diversification in Japan within the past ten years. This rapid increase in diversity comes with the formation of new social alignments and the challenges of "the Different" [4]. Though Japan is diversifying, this is not visibly seen throughout the country; "patches of visibly diverse districts", which Tsuneyoshi refers to as "diversity points" [4], are "scattered amidst a vast sea of seeming homogeneity" (p. 57). Maher argues that "the notion of contemporary Japan as essentially 'multicultural' hits a raw nerve and becomes radically controversial when we turn to official descriptions" [20], and that the idea of a diverse Japan is "systematically avoided, above all in the post-war period, by the so-called Nihonjinron theories, and the ethnocentric body of writing which emphasizes the uniqueness of 'being Japanese' (e.g., Doi's Anatomy of Dependence, Nakane's Society, Lebra's Japanese Patterns of Behaviour)" (p. 125).

Table 1 provides a breakdown of foreign nationals in Prefecture M. in the years 2014–2018. Over 70% of the new generation of immigrant children originate from neighboring Asian countries such as China, Korea, Vietnam, the Philippines, Nepal, and Indonesia. Like their immigrant parents, they are from highly diverse socioeconomic backgrounds. Zhou and Bankston state [3], "Their economic situations, educational attainment and health will shape their own future while significantly influencing the futures of their host countries" (p. 1). In Japan, minority populations, which were once seen as "invisible" with the majority being from East Asian countries (such as China and Korea), are now becoming visible with increased ethnic diversity being more evident in many communities [21] (p. xix). Japanese citizenship laws reflect the strong value placed on kinship-based blood ties, and Japanese policies have often emphasized this by excluding cultural minority groups [4,5]. Though predominately Asian, some groups of "new" foreigners tend to stand out visibly compared with foreigners who have lived in Japan for a longer time. They speak a different everyday language, have foreign names and foreign customs, and some look different [4]. Japan is seeing a shift from an "invisible" diversity—for example, many Korean and Chinese nationals in Japan were born and raised in Japan, have adopted Japanese names, and speak flawless Japanese—to a more "visible" diversity of people coming to Japan as foreign

laborers, refugees, and spouses of Japanese. Newcomers include (but are not limited to) individuals ranging from adult blue-collar labourers to academic researchers to highly skilled professionals.

**Table 1.** Statistical Information on the Foreign resident population in Prefecture M. [22].

| Year | 2017 | 2018 | 2019 | 2020 | 2021 |
|---|---|---|---|---|---|
| Korean/North Korea | 25,962 | 25,455 | 24,909 | 23,809 | 22,830 |
| China | 15,585 | 16,939 | 18,504 | 17,037 | 15,248 |
| Vietnam | 3246 | 4506 | 6130 | 6667 | 6438 |
| Philippines | 2242 | 2368 | 2435 | 2410 | 2429 |
| USA | 1430 | 1521 | 1591 | 1438 | 1414 |
| Indonesia | 838 | 905 | 1051 | 1036 | 946 |
| Nepal | 635 | 759 | 1052 | 1044 | 983 |
| Thailand | 626 | 737 | 760 | 716 | 634 |
| France | 603 | 651 | 693 | 586 | 535 |
| Other | 5311 | 5791 | 6395 | 5905 | 5679 |
| TOTAL | 56,478 | 59,632 | 63,520 | 60,648 | 57,136 |

In understanding different minority groups and the reasons why acculturation (which may impact children both socially and psychologically) and school learning are challenging for some, it is useful to refer to Ogbu [23], who categorizes minority groups as (a) autonomous (people who are minorities in a numerical sense); (b) immigrant or voluntary (people who have moved more or less voluntarily due to pull factors such as better socio-economic opportunities or political freedom); or (c) involuntary minorities (people who are forced to leave their country against their will). Individuals in category (b) (immigrant or voluntary) are one group of minorities who are included in this study. Ogbu states that voluntary minorities usually experience initial problems in school due to cultural and language differences as well as a lack of understanding of how the education system works (p. 363).

The term foreign nationals is used to refer to people who do not have Japanese nationality (Immigration Control and Refugee Recognition Act, 2014, cited by [24] p. 11). The Ministry of Education, Culture, Sports, Science and Technology (hereinafter referred to as MEXT) acknowledges the need for Japanese language assistance for foreign-national students due to the rapid expansion of the foreign population. Government statistics show that foreign children who require Japanese language assistance accounted for approximately 0.15% of the total enrollment of students in primary and secondary schools in 2015 [25]. The latest statistics show that as of 2021, more than 47,000 children are in need of additional Japanese language support ([26], https://www.e-stat.go.jp/ accessed on 12 August 2023).

*2.1. Terminology Used to Understand Ethnicity and Identity in Japan*

Murphy-Shigematsu et al. divide ethnic minority children in Japan into various categories [21], including (1) children who hold more than one passport with dual or multiple nationalities, (2) children who were born in Japan but legally are a national of another country or were born in a country other than Japan, and (3) nikkeijin (non-Japanese of Japanese descent) of South America (p. 136). The Immigration Control and Refugee Recognition Act was revised in 1990 to allow younger generation nikkeijin to take unskilled work legally in Japan. This resulted in a visible increase in the number of South Americans and other identifiable types of foreign nationals, such as foreign spouses of Japanese and children of international marriages. Children who are of mixed ethnic or national background, more particularly Eurasian children, even if they are Japanese nationals, are popularly marked out by the terms hāfu (half) or dāburu (double). Widely used and not

necessarily pejorative, these appellations are a reminder that there is one kind of Japanese and an "other" kind [20]. Okano explains that terminology has emerged in titling ethnic differences based on government policies [19], with the term "foreign nationals" in Japan referring mainly to Korean nationals. It was only in the 1980s that foreigners who were not Korean nationals began to be taken notice of. Given that the multicultural education policies were initiated for the education of Korean residents, the government was obliged to acknowledge in the policies Korean residents' desire that their identity be recognized as distinct from that of new immigrants.

The use of these limited terms to represent ethnic identity in Japan is problematic, as they exclude Japanese citizens of non-Japanese descent: those who have taken up Japanese citizenship, and children of Koreans and/or foreigners who are granted Japanese citizenship at birth. Once a foreign national takes up Japanese citizenship, he or she simply disappears into the Japanese citizen category at that point, and his or her ethnic background ceases to exist in the official discourse. While officially they have ceased to be foreign nationals, their ethnic identity and sense of marginalization may continue to exist. The terminology excludes indigenous peoples (Okinawans and Ainu). There are no official documents detailing the ethnic composition of the Japanese population nationwide, in contrast to, for example, a country like Australia, which collects such information in each national census. A few local government policies have started to recognize the diverse ethnic ancestry of Japanese citizens in the student population and recommend that the same policies be applied to Japanese nationals of non-Japanese descent. Daily interactions with Korean nationals of Japan suggest that the understanding of citizenship among Japanese youth has begun to be more inclusive [19].

### 2.2. Definitions of the Ethnic Minority Children Who Are the Focus of This Study

The internationalization of education has become a trend in Japanese national policy, as evidenced by the inclusion of courses of studies in "education for international understanding" in the national curriculum. This may be partly in response to the increasing globalization of Japanese society, with growing numbers of foreign residents and ethnic diversity of students becoming more visible in schools throughout Japan. Ethnic minority children (hereinafter referred to as EMC) are defined and categorized in many ways in Japan based on their unique situations, linguistic and cultural backgrounds, and specific language learning needs. They are also categorized for reasons related to creating policies and systems. Though policies have been put in place in an attempt to accommodate their social and academic needs, many EMC constantly exist "in between" their Japanese and their other ethnic culture(s). Society tends to mark and treat these children as "other", which at times has negative connotations in Japan.

Figure 1 depicts the two categories of ethnic minority children being focused upon for this study.

The first category is primary school children who were born in a foreign country, whose parents are both of foreign nationality and who have recently migrated to Japan. Japanese is their second language. This group of children are commonly referred to by a variety of terms, including gaikokujin no kodomo or gaikokujin jidō seito (foreign children or foreign school children); gaikoku shusshin no jidō seito or kikoku tonichi jidō seito (school children of foreign origin or returnee school children); and nyūkamā no kodomo-tachi (newcomer children) [27–30].

The second category is primary school children who are Japanese nationals, were born from international marriages (in many cases, one parent is of Japanese nationality and the other of foreign nationality), and have been raised in Japan. Though many of them are bilingual and bicultural, and Japanese may not be their dominant language or culture. This group of children is most referred to using the terms gaikoku ni tsunagaru kodomo, gaikoku ni roots o motsu kodomo (children with foreign roots abroad), or idō suru kodomo-tachi (migrant children/children crossing borders) [27,31–33].

**GAIKOKUJIN JIDOUSEITO**
(*Foreign school children*)

●school children who were born in a foreign country, whose parents are both of foreign nationality and who have recently migrated to Japan.

●children who are not familiar with Japanese language or Japanese culture

●whose mother tongue is a language other than Japanese

●who require Japanese language support at school

*Other terms researchers have used to describe this category of children:*

'gaikokujin no kodomo'
(*'foreign children'*)

'gaikoku shusshin no jidouseito'
(*children of foreign origin*')

'nyucama no kodomo'
(*'newcomer children'*)

'kikoku tounichi jidouseito'
(*'children of foreign origin'*)

**GAIKOKU NI ROOTS O MOTSU KODOMO**
(*Children with ethnic roots abroad*)

●school children – Japanese nationals - who were born from international marriages (in most cases where one parent is a Japanese national and one parent is a foreign national).

●who have been raised in Japan or partly raised in Japan

●are both bilingual and bicultural, and may speak a language other than Japanese at home

●Japanese may not be their dominant language and may require Japanese language support at school

*Other terms researchers have used to describe this category of children:*

'gaikoku ni tsunagaru kodomo'
(*'children connected to foreign countries'*)

'idou suru kodomo'
(*'migrant children' or 'children crossing borders'*)

**Figure 1.** The Two Categories of Ethnic Minority Children Who are the Focus of This Study.

## 3. Methodology

This is an exploratory study that seeks to examine the experiences and perceptions of the provision and practice of Japanese language support for ethnic minority school children aged between 6 to 12 years of age currently enrolled in public elementary schools in a city located in the Kansai region (hereafter referred to as 'City M'). The research design for this study follows a qualitative approach being exploratory, descriptive and predictive in nature. This study adopted qualitative methods for data collection and analysis. The researcher preferred to gather empirical data rather than statistics or measurements, in the attempt to examine phenomena that impact the lives of individuals and groups of a particular cultural and social context. She placed a considerable amount of value on the understanding of the individual voices and experiences of her participants.

### 3.1. Reasoning for Choosing Qualitative Research for This Study

According to Strauss and Corbin [34], qualitative research is 'any kind of research that produces findings not arrived at by means of statistical procedures or other means of quantification. It can refer to research about persons' lives, stories, behavior, but also about organizational functioning, social movements, or interactional relationships (p. 17). Qualitative research can be understood as 'a research strategy that usually emphasizes words rather than quantification in the collection and analysis of data' [35] (p. 11). The research design for this study follows a qualitative approach being exploratory and descriptive in nature.

### 3.2. Semi-Structured Interviews

A semi-structured interview is a type of interview used in qualitative data collection in which the researcher asks informants a series of predetermined but open-ended questions and creates a written interview guide in advance [35]. Semi-structured Interviews are used to gather focused, qualitative textual data, having the flexibility of an open-ended interview and the focus of a structured ethnographic survey [36]. The interviews were

audio-recorded and then transcribed. Collected data was coded and analyzed thematically with the use of NVivo qualitative data analysis software.

The researcher conducted a thematic analysis, in which data from the interviews was examined to extract core themes that can be recognized both between and within interview transcripts [35] (p. 12). The identification of themes was done through coding each transcript, in which pieces of data will be categorized into component parts and given specific labels. The researcher then searched for similarities and recurrences in the different sequences of text in the data and looked for links between different codes. The researcher made sense of the data by coding the interview transcripts and examining and determining relationships between the data. Trying to make sense of the information with reference to the research questions, various government-created documents as well as past and current literature on the research subject was also conducted. The collected data was classified and organized into first identifiers and then concepts. Concepts will then be divided into categories, with further being reduced down to core categories, which will become the basis for the grounded theory.

### 3.3. Participant Recruitment

Interviews were conducted both individually (one-to-one in-depth interviews) and in pairs and groups (interviewing two or more participants at the same time). The length of the time of the interviews ranged from 30 to 60 min. Depending on the availability of the participants, the place of the interviews ranged: school classrooms, principal's offices, government offices, lobbies of public spaces—wherever the participant felt comfortable talking.

### 3.4. Data Collection Method

By using the method of qualitative data collection, this study aims to capture the personal and individual experiences of participants who may be able to widely represent many of their peers on some pressing and significant issues. Various data collection techniques were used by the researcher such as purposive and snowball sampling. Purposive sampling is a way of sampling qualitative research that involves selecting participants or data on the basis that they will have particular and distinct characteristics or experiences [37] (p. 56). Snowball sampling proved very effective in the recruitment of participants, possibly because many of the participants wished for their words to be documented; for their personal experiences and perspectives to be recorded, rather than their voices just being summarized and coded into a numerical statistic. As for data collection, information was gathered from the interview transcripts as well as a collection of archival data in order to secure and assure multiple perspectives and various levels of analysis.

### 3.5. Nvivo Qualitative Analysis Software

Used as a tool to help categorize and organize information in the interviews, Nvivo, a qualitative analysis software program was used to organize and code common themes and help with the analysis of data. Data was organized into codes or "nodes", as this analysis software terms it, in order to find similar concepts and categories, with the goal to finally develop some possible theories which emerge from the data.

### 3.6. Positionality

The researcher of this study considers herself an "insider" in being a foreigner and a mother with a primary school child categorized as an ethnic minority child in City M. The researcher feels that she may have different perspectives on certain issues from those of a mainstream Japanese national. She, therefore, feels that though the interpreted results and discussion presented in this paper may be biased to a certain extent, she hopes that the bias will add a much-needed, 'insider' perspective which has been seldom voiced in the current literature. Also, the researcher was closely involved with various local volunteer groups in City M and other associations within the international community who are advocating for support of ethnic minority children and their social well-being and educational needs.

Because of her active involvement, beliefs and opinions may seem disproportionate in weight in favor of the volunteer language support groups.

*3.7. Participants*

The group of participant interviews to be analyzed in this paper includes the following members (See Table 2 below):

1.  In-house Japanese language support teacher staff at public schools Teachers employed by the City M Board of Education to specifically provide Japanese language support to children who require support in Japanese. Depending on the school, the support teacher provides support in meeting their specific language needs (from basic "survival Japanese" level to "elementary Japanese" level) [25] (pp. 27–28). In-house Japanese language support teachers work with the child, either on a one-on-one basis, or teach a group of children in a designated class for Japanese language support. The backgrounds of the two teachers who participated in this research project vary (based on their years of JSL teaching experience and specialized skills in teaching Japanese to EMC requiring Japanese language support.

2.  Volunteer interpreters Registered volunteers who are dispatched by City M Board of Education. These volunteers provide interpreting services for children and their parents at public schools (interpreting either in the mother tongue of the child or a language most familiar to them). The volunteer's years of experience in providing these services vary between 5 to 30 years. Volunteer interpreters support EMC who require assistance in understanding what is being taught in the classroom and well as support with comprehending the instructions of the teachers, and textbook explanations and assigned homework [25] (p. 60).

3.  Mother tongue language supporter (hereafter referred to as "MTLS") Registered staff dispatched by City M Board of Education to provide services mother tongue language support to children at public schools. Mother tongue language supporters assist EMC who do not possess a sufficient level of Japanese language ability and require assistance in their mother tongue in order to understand what is being taught in the classroom (in particular, students who have just come to Japan recently and have newly enrolled in the school.) This support is usually one-on-one support with the student [25] (p. 60).

4.  Principals of public elementary schools in City M Principals from three different public elementary schools in City M in different regions of the city. Each school has unique demographics, school cultural backgrounds and varied levels of experience in providing learning support at their schools.

**Table 2.** List of 9 participants who are categorized under language support teacher (employed at public school/employed by Board of Education, volunteer interpreters, mother-tongue language supporters (dispatched by BoE), and senior management at public schools in City M.

| Participant No. | Job Title | Interviewee Category | (1)<br>(2) | Job Description/Role<br>Years of Experience |
|---|---|---|---|---|
| #30 | Japanese language support teacher at Public elementary school N | (g) Japanese support teacher employed by City Board of Education | (1)<br><br><br>(2) | Teaching Japanese language for ethnic minority students who require additional support<br>5 years |
| #35 | Principal at Public Elementary School Principal A | (h) Public Elementary School | (1)<br>(2) | Elementary school Principal<br>28 years as an educator |
| #37 | Principal at Public Elementary School Principal I | (h) Public Elementary School | (1)<br>(2) | Elementary school Principal<br>20 years as an educator |

**Table 2.** *Cont.*

| Participant No. | Job Title | Interviewee Category | (1)(2) | Job Description/Role<br>Years of Experience |
|---|---|---|---|---|
| #38 | Principal at Public Elementary School Principal B | (h) Public Elementary School | (1)<br>(2) | Elementary school Principal<br>32 years as an educator |
| #39 | Japanese language support teacher at Public Elementary School A | (g) Japanese support teacher employed by City Board of Education | (1)<br><br><br>(2) | Teaching Japanese language for ethnic minority students who require additional support<br>20 years as a JSL teacher at an international school abroad and at Japanese public schools |
| #2 | Volunteer Interpreter/MTLS (Mother-Tongue Language Support) | City Board of Education | (1)<br><br><br><br>(2) | Dispatched by City M BoE to provide interpreting services and/or mother-tongue language support for school children and/or their parents<br>5 years |
| #1 | Volunteer Interpreter/MTLS (Mother-Tongue Language Support) | City Board of Education | (1)<br><br><br><br>(2) | Dispatched by City M BoE to provide interpreting services and/or mother-tongue language support for school children and/or their parents<br>10 years |
| #18 | Volunteer Interpreter/MTLS (Mother-Tongue Language Support) | City Board of Education | (1)<br><br><br><br>(2) | Dispatched by City M BoE to provide interpreting services and/or mother-tongue language support for school children and/or their parents<br>15 years |
| #41 | Interpreter (also a Japanese language support teacher for BoE) | City Board of Education | (1)<br><br><br>(2) | Dispatched by City M BoE to provide interpreting services for school children and/or their parent<br>8 years |

It should be noted that some codes do appear to overlap in description—this is because many of the statements made in the interviews may have been allocated to several codes. Table 3 shows the text of the indicators taken coded from the initial interview analyses and based on saturation of the indicators and overlapping themes, were then categorized into concepts. It also must be noted that some of the participants interviewed held more than one type of job title such as participants #1, #2 and #41.

### 3.8. Indicators and Concepts

From my interviews with Japanese learning support teachers and language support staff, I found common threads in their conversations. The common threads are noted below as indicators. I have compiled the indicators and from their content have examined their interconnecting meanings and have developed concepts representing their meanings.

It should be noted that some indicators may appear to overlap in description—this is because many of the statements made in the interviews may have been allocated to several indicators. Table 2 shows the text of the indicators coded from interview analyses and based on saturation of the indicators and overlapping themes, were then categorized into concepts. It also must be noted that some of the participants (such as participants #1, #2 and #41) held more than one type of job title and therefore have been included in other participant groups discussed in other chapters of this research paper.

**Table 3.** List of Indicators and developed concepts.

| Identifiers (Participants) | Indicators | Concepts |
|---|---|---|
| #1, #2, #18, #30, #29, #41 | providing language support is much more than just teaching the language | Supporting EMC at schools |
| #1, #2, #18, #30, #29, #41 | many different expectations of roles and responsibilities | |
| #1, #2, #18, #39, #41 | Infrequency of support at schools | |
| #1, #2, #18, #39, #41 | feel the need to act as an academic advisor for parents | |
| #1, #2, #18, #39, #41 | acting as a child counsellor for students | |
| #2, #18, #30, #39, #41 | Need for a support network to exchange knowhow and expertise with fellow language supporters at other schools | |
| #1, #2, #18, #41 | not feeling heard nor supported by school or board of education | Lack of effective communication between language supporters, school and BoE |
| #1, #2, #18, #39, #41 | feel that they are the only voice to represent children and their parents when communicating with school | |
| #1, #2, #18, #37, #39, #41 | Teachers at the school don't have enough knowledge or know how in interacting or supporting EMC | |
| #2, #37 | Schools do not know how to interact with EMC | Schools' lacking knowhow and expertise in interacting with and supporting EMC and their families |
| #1, #2, #39 | Feelings of ignorance and apathy within school community about ethnic minority children learning needs | |
| #1, #2, #18, #30, #37, #39, #41 | Lack of access to resources to receive mother-tongue language support for languages other than English | |

*3.9. From Concepts to Categories*

From the indicators and abstract concepts, I created 5 categories in order to start the process of analysis. In the following sections, I will provide a processual narration for each of the categories, examining and analyzing excerpts of interviews that express themes embedded in the categories.

- Quality, quantity and varieties of learning support needed
- Expertise and know how in supporting EMC learning needs
- Disconnect between EMC, parents, support staff and school
- Review and revision of current policy and system
- Training for in-service teachers

*3.10. Quality, Quantity and Varieties of Learning Support Needed*

I conducted interviews with school leadership staff and Japanese support teachers at three different public elementary schools in City M. Depending on the school's historical background, experience in providing language support and overall expertise and know-how, there is a disparity in the quality and quantity of language support offered at public schools.

Participant #30 is a Japanese support teacher employed at "School N", a public school in an area of the city near a national university where many academics from abroad come to study at graduate school or to conduct research. It is in this area where many of these foreign researchers reside together with their families. Many of the foreign researcher's children are enrolled at School N. This particular public school is known not only for the number or EMCs but also as a school that provides sufficient language support for foreign children who do not have any prior knowledge or skills in the Japanese language or culture. According to Participant #30, depending on the time of the year, an average of 5–10 foreign

children are enrolled at this school annually, and they usually stay at the school between 1–3 years.

Memo: 14 July 2018

When I visited this school for the first time, I noticed signage around the hallways (for example, the bathroom, or the teacher staff room) written in different languages (Chinese, Korean and English), and learned that it was not the teachers, but the students themselves who made the signs. Participant #30 recalls that the younger students created and posted these signs for a project some of the classes were taking part in a few years ago but could not recall about the specific purpose. (14 July 2018)

Participant #30 talks about the class that she is responsible for teaching and how it supports their ethnic minority students:

*I am teaching a special class called Nihongo Gakkyu (Japanese language studies) where I teach first-time basic Japanese (shoki Nihongo) to students; teaching style ranges from teaching small groups of children to one-on-one instruction. I also teach children who are pulled out from their regular classes to join in this special class. Nihongo Gakkyu class is held in the International Classroom (hereinafter referred to as "IC"). We hold this class 3 to 4 times a week* (Participant #30).

Participant #30 talks about being the Japanese support teacher at the school:

*I'm not just teaching the language to these children; since many of them do not know nor understand Japanese culture or its rule, it is also my role to teach them about culture, common school rules and many things other than just language. Therefore, there are a lot of things that I need to check up on, investigate and inquire when working with these children* (Participant #30).

Language support teachers find themselves not just supporting the students in language instruction. They are also teaching them about visible culture, such as daily Japanese etiquette and customs, as well as invisible culture, such as unspoken rules in daily school life and society. This additional role takes increased preparation time on top of preparing the Japanese language class content.

Participant #41 one of the younger language supporters within this group of participants, talks about what she does in supporting those newly enrolled EMC who do not understand the Japanese language:

*I sit beside them in class and try to help them understand what is going on in the class, what the teacher is saying what the other classmates are talking about. I try to explain to the student the teachers' instructions and their explanations in the textbook. However, I do not know how to clearly explain about the lesson. It is just not enough to translate solely from Japanese into English. The job and role of the volunteer translator is also to explain the cultural background behind the meanings* (Participant #41).

Despite School N having a low number of foreign children needing language support, this school has made the efforts to create an international classroom (IC Room) with a full-time support teacher on duty, as well as employing a staff member to do translation of school documents for parents into English, as well as interpreting services for parents. The reason for this school's extensive efforts may be because of the academic status of the children's parents. Having adequate educational services for their children might be an incentive for hailing world-renowned academics and researchers to come to the university. Beneficial for the reputation of the university and for the city itself in its aims to become known as a growing hub for top academic researchers.

Participant #18 is a mother-tongue language support staff and interpreter volunteer in the city. The participant explains what their roles and duties are:

*Not only do I have the roles of mother-tongue language supporter and volunteer interpreter, I also feel that my part of my role is to try to help the children process with their*

*feelings of anxiety and insecurity about being an ethnic minority in Japan. I am a source of reassurance to them; a kind of counsellor providing psychological care to these children and their parents (Participant #18).*

### 3.11. Determining the Quality and Quantity of Learning Support

Participant #37 is a principal at "School O" which was established in City M approximately 88 years ago. I learned from the participant that the school is located in an upper-class neighborhood where 30 to 40 percent of the parents of the school are highly skilled professionals (such as doctors, and university professors). When asked about the EMC enrollment at the school, the participant replied that while on average there is at least one student per class who is identified as an ethnic minority student, they may not especially require additional Japanese language support.

*Up until now, we did not need to request Japanese language support from the Board of Education, but recently, we have had a few cases where children enroll for a short period to our school and need language support. It at this time that we started to request a couple of times in the past year. The support teacher usually comes 2–3 times a week, but since the students' Japanese ability seemed to be improving (as children acquire a foreign language very easily. . .), we decided to decrease the number of times for the support teacher to come to the school. We believed that the support was no longer needed so frequently (Participant #30).*

When asked whether they felt that the child would have benefitted more by receiving more language support, the participant was quick to say that the school decided that offering more support was no longer necessary, since the student "seemed to be getting along well". It is curious to ask how the principal and the homeroom teacher came to the conclusion that further language support was no longer needed. What benchmarks or systems were in place that were passed by to let the leadership team make the decision to decrease the amount of time the child was receiving learning support? What authority or expertise does the school leadership team have to make the decisions of learning support for foreign children at their school?

### 3.12. Expertise and Know Know-How in Supporting EMC Learning Needs

Participant #39 is a language support teacher with over 30 years of experience in providing Japanese language support both in Japan and abroad. She was currently employed by the city's board of education. She explains her job and role as a language support teacher at a particular city public elementary school.

*I work at five different public elementary schools, four which are primary level and one school that is junior high level. I work with many children from many different backgrounds: those who cannot speak any Japanese and require first year beginner language instruction, as well as children who were born in Japan but require Japanese language support. . ..their nationalities are various, both Japanese and foreign children (Participant #39).*

When asked about their role as a Japanese language support teacher, Participant #39 explains her concerns about her own abilities and credentials in being able to effectively help students in their learning and helping them attain their educational goals:

*I think that role of the support teacher is to help the student be able to understand enough Japanese to be able to follow what is going on in class. However, since I am not the homeroom teacher, I don't know in details what the exact learning outcomes for the student in that grade should be and therefore am not clear about what particular goals or aims the student should accomplish in terms of content in the subjects for their particular grade level. If the support teacher does not have experience in teaching at a public school, it is very difficult to provide support for the student if the support teacher doesn't know what is expected of the student in the classroom in terms of the required level of learning and understanding according to each subject (Participant #39).*

Participant #39 voices her concerns about whether a support teacher is really able to understand and determine the learning needs of the student she is working with. She is questioning her expertise as an educator who is responsible for not only providing support for the students but also increasing their levels in Japanese language not only to understand Japanese but to be able to help improve and enrich their learning abilities and performance at school.

*3.13. Supporting EMC with a Mother-Tongue Other than English*

*Teaching and supporting EMC whose mother-tongue is a language other than English is the most challenging thing in my job* (Participant #30).

*There is little information and learning support materials available in different languages other than English and Chinese. For example, there is little information and materials in Spanish available* (Participant #18).

There are more in-service teachers at public elementary schools in Japan with a sufficient level of English who can help support some EMC who can either communicate or are familiar with English and are able to be supported in this language. However, with an increase of foreign children with mother tongue other than English, it is proving more difficult for learning and language support staff to access resources to help meet the growing need for support in minority languages.

Participant #37 explains:

*If the child's mother tongue is English, it is not such a problem to support the student as we have teachers who can speak and understand English. However, if the child's mother tongue is a language other than English and the child (and parent) does not understand Japanese, it is very difficult to communicate with the child and their parents, so we rely on the interpreting supporters (university student volunteers dispatched by the city's international exchange foundation) for assistance* (Participant #37).

In many cases similar to the one above when the student's mother tongue is a minority language in which there is no volunteer translator available or accessible, it often happens that two volunteers are assigned to the student: one language supporter who assists the child in the mother tongue and then translating into English; the other supporter translating from the English to Japanese, and then visa versa. In the event that the school or Board of Education cannot find anyone to assist the child in their mother tongue, the volunteer supporter tries to communicate with the child in English, which may be a language not so familiar or accessible to the child.

*We are having a very difficult time trying to find someone who can speak Arabic. We have tried to ask the Board of Education for help, they could not find anyone. We have a particular student who requires this mother tongue language support and the child needs help. The Japanese support teacher is currently trying to help the child in their broken English* (Participant #37).

When the participant was explaining about this student, it seemed that the child was in a lower grade of primary school and did not understand or speak English very well. This means that even if the support teacher tried to communicate in English with the child, no effective communication could come out of their efforts, as the child could not even communicate or comprehend English. This exemplifies both the stigma and misconception that all foreign children can communicate English, regardless of age. It cannot be assumed nor expected that young children, who are still in the middle of developing their linguistic abilities and acquiring their mother tongue have high proficiency in the English language.

MEMO: 2 December 2018

I was surprised that "School O" failed to find support or even at least access a network with an organization that would be able to offer support to the child in their mother tongue. I also wondered why the school did not pursue other venues to find information or actively reach out to international exchange agencies,

cultural groups or other educational institutions to search for language support. After the interview, I offered to share my network with the participants in order for them to access a grassroots organization that could offer her some interpreting support for the child. Had I not volunteered to offer my assistance in accessing language support for this student, I wonder whether or not the school would have remained inactive to find someone.

### 3.14. Concerns about Official and Unofficial Duties of Support Volunteer

In regard to the duties of volunteer interpreter and mother-tongue language supporter, the participants who shared their experiences with me talked about the duties they are expected to fulfill, boundaries that they cannot help to cross and the unknown needs of the ethnic minority families.

Participant #2 explains below her frustrations about where to draw the line regarding the duties of a support volunteer:

> *Though we are told strictly to not get too involved with the students and their families (don't exchange personal information, don't support them too much, etc.), there are sometimes when the students and their families desperately require more help than just language, and us volunteers are at a loss as to what to do to help them. Our hands are tied and we are told not to help them by the city, but they need our help* (Participant #2).

Though establishing clearer guidelines about the roles and responsibilities of the support volunteers may help to streamline their duties to make it clearer for the volunteer to understand, it may, on the other hand, disable the support the volunteers can actually offer those individuals who are seeking help and have no other lifeline to access.

### 3.15. Preparing and Supporting Students to Enter Secondary Education

When asked what the challenges of teaching students in IC were, Participant #30 responded:

> *In the higher grades such as grades 4, 5 and 6, the level of the study content in these grades becomes more difficult, in particular, Kanji (Chinese character writing). The level of study becomes more difficult for students in these grades. For students who will soon return to their home countries, it is not that important for them to achieve well in the classes. However, for those children who plan to go onto junior high school in Japan, advancing to that stage is very difficult* (Participant #30).

Language support teachers supporting children at the primary school level may be able to help these children with their studies while in the younger grades. However, it may be questioned as to how confident the teachers are in being able to support EMC students in the latter half of their primary school education in preparing them for advancing to the next level of education in Japan. They might not have the expertise or know-how to help some foreign children, who have different educational needs than mainstream Japanese children in helping them study for entrance examinations for junior high school. For many of these teachers, they are at a loss about how to help these students with their future education, with no one to guide them or direct them to help or assistance.

### 3.16. The Disconnect: EMC, Parents, Support Staff and School

Scholars, educators and families have frequently indicated that communication between parents, schools and teachers is one of the most important ingredients required for the academic success of children [38]. Parental communication with teachers, who are typically the school faculty most familiar with the child, can help parents better track the academic progress of their children. Moreover, teachers can also better understand emotional developments at home that may shape how the child is learning in the classroom [39,40].

*3.17. Communication between School and BoE*

Participant #2 has been a volunteer interpreter and mother-tongue language supporter for several years in City M. This participant has seen the changes in policy and systems of learning support throughout the years and shares personal thoughts about why EMCs are still not receiving the support they need at schools.

> *One of the reasons why some EMCs are not receiving the support that they need is that their schools are not providing the services. The school must make an official request to the BoE to access these services. If the school does not show any concern or make the effort to make a request for the child, then the child is unable to get access to these support services* (Participant #2).

The participant feels that the reason is due to the disconnect of the different parties responsible for providing the support:

> *".... some teachers may see it as "more work for them to do", especially those who are already overworked" or it may be seen as a "burden" for the school to request such services from the BoE. There are more public schools out there who are making the efforts to request the services needed for their students, but there are still many families who do not know that such language support services are provided by (City M) because their schools do not provide any information. BoE is waiting for the requests from schools* (Participant #2).

When asked the reasons why schools are not so active in requesting support for students, the participant replies that it may be related to the teachers' overwhelming workload at the schools. It is important to note here that in Japan teachers do not select the schools in which they teach; rather they are assigned by their prefectural Board of Education [41]. Moreover, teachers are rotated regular basis, usually every 6–7 years. The situation is more extreme for principals, who serve for only 3 years at a school. It is argued that the rotation policy educators an opportunity to work in a variety of settings and to grow professionally. However, in reality, this policy does not ensure the continuation and clear transmission of knowledge, expertise and know-how in interacting and supporting ethnic minority children with their learning at schools.

*3.18. Communication between Support Teacher and the School*

Participant #39 talks about documentation writing and maintaining communication with the teachers at the school:

> *Support teachers are given specific instructions by the Board of Education of fill out reports to document the activities at the end of each session with a student. We have to write down in the limited space that is provided on the form of the report what we did in the session, what the student did and the teacher's observations of the student. This form is then supposed to be read and checked by both the homeroom teacher and the head teacher responsible. Once they read the report, they can then have a better understanding of the student, know to how to deal with the student and better support them in their learning. I then try to meet and talk with both the student and homeroom teacher to maintain good communication with both parties. Once the student has increased their language proficiency enough to be able to follow the class lesson, I try to follow up with the homeroom and head teacher to ask what things I can do with them in to increase the student's confidence in their learning in the classroom. However, since the homeroom teachers are so busy, I don't bother to ask them too much* (Participant #39).

While there are systems in place for report documentation about the activities of learning support with the child, as for the transmission and exchange of information, as well as direct communication with the homeroom teacher or leadership team of the school, there are problems in how information is communicated and documented.

*3.19. Disconnect between Support Volunteers and BoE*

When talking with the support volunteers, they expressed their frustration with communicating with the BoE regarding matters such as their roles, duties, responsibilities and transportation payment. All of the support volunteers felt a communication gap between them and the BoE who dispatched them to schools.

> *"The BoE has no idea what the volunteers experience or what they are asked help with from the families and children, because they are not there at the schools talking with these people. The City and BoE need to talk to both the schools and the EMC parents and have some sort of information exchange in order to figure out what measures to take and what to do to help these children and their families. The BoE tells the volunteers "you don't have to do that much" or "you don't have to provide that much support ", but if volunteers do not do these things, nothing will be solved for these children or families".* (Participant #2)

The supporter above expresses her frustration being a long-time support volunteer for the city, mainly because the roles of the job are not clearly defined. Though there are limits and restrictions about what a supporter volunteer can and cannot do, they feel that there are unspoken duties that they are expected to fulfill. In addition, with such a small ethnic minority community with limited resources and support networks, volunteers also may feel that they have a humanitarian duty to do more than expected by the BoE for the sake for the sake of the child and his/her family's social and mental wellbeing.

*3.20. Disconnect between School and EMC Parent*

> As the worries and concerns that Japanese families are very different than ethnic minority families; concerns, insecurities, aims and hopes for the future, so it is difficult for homeroom teachers to provide learning support needed. (Participant #18)

Participant #18 argues that parents don't have enough awareness about Japanese language support which their children require. She also claims that there is not enough support provided for families in the local communities and at the municipal level.

> *...I 'm having myself speak up for many EMC parents communicate with the BoE or City on their behalf because they do not feel comfortable speaking to their child's teachers due to language barriers, communication issues with the teachers. They feel that they do not have a voice in their child's education at the public school.* (Participant #2)

Reliance on interpreting services provided by the Board of Education and international centers in City M do help break down the language barriers between parents and schools. However, it cannot be seen as an all-end solution to narrowing the communication gap between EMC parents and schools. Senior leadership at schools must implement more of a sustainable, self-reliant solution for schools to be able to foster communication with EMC parents and support them with their children's learning. For example, the school can make active efforts to seek advice and guidance from local support organizations (international exchange associations, NGO/NPO groups, universities, etc.) and invest in supplementary teaching materials).

*3.21. Review and Revision of Current Language Learning Support System in City Demographics*

Despite increasing numbers of EMCs throughout the prefecture and city, the numbers are scattered throughout each region, with only a few areas where groups of ethnic minorities tend to reside in one area. Therefore, it is especially difficult for public schools to justify the establishment of a Japanese language support classroom. Unlike Osaka and Tokyo, where there are a number of ethnic minority communities, and thus more obvious to understand the need for a classroom, looking at the current demographics and numbers in City M are not substantial enough to convince more progressive action.

*I think that there are a limited number of Japanese language support volunteers for two reasons; one being that the schools in which they are needed are spread all throughout the prefecture, so it is very difficult to travel to these schools that are so far away and difficult to access by public transportation.* (Participant #2)

When asked what they thought was needed to improve the provision and practice of Japanese language support at public schools all participants who are focused on within this chapter (Participants #1, 2, 18, 41, 30, 35, 38, 39, 41) suggested the following:

1. Increase the number of hours for Japanese language support and the number of Japanese language support teachers at schools. (All participants)
2. Create a system where there is at least one teacher with expertise in Japanese language support and understanding of the various learning needs of EMC. (Participant #2, 18, 39)
3. Increase the number of teachers at public schools who can speak either a mother-tongue of the students or at least a second language. (Participants#2, 18, 39)
4. Increase the budget in order to be able to provide support not only for the child, but also for their families. (Participant #2, 18)
5. The Board of Education must make clearer, more attainable guidelines for the roles of language support volunteers. (Participant #1, 2, 18, 39)
6. Increase the amount of financial support for the support volunteers. (Participant #1, 2, 41)
7. There should also be opportunities for language support volunteers to discuss together their experiences and chances for information exchange in order to open dialogue, share concerns, etc. (Participants #1, 2, 39, 41)

Participant #35 is one of the head educators at a primary school with over a 300-year history in the city, a school that existed during the Meiji period. The participant voiced the need for the creation of a Japanese learning support system that each school can use effectively.

*For example, when there is an increase of EMC, schools should be able to receive the support and permission to set up a Japanese language class or special Japanese language classroom. They (BoE) should increase the number of teachers who are specialized in Japanese language support to be dispatched at schools. If there are more teachers who can be dispatched at the schools, more EMC an receive the proper type of support which they need, as well as have the sufficient amount of time to receive this help. The amount of hours for providing this support to students will then increase and therefore will be able to provide more quality support* (Participant #35).

### 3.22. Training for Pre-Service and In-Service Teachers

In preparing for an influx of ethnic minorities in the city, schools and local communities need to understand about ethnic diversity and create a community that includes and welcomes diverse individuals. One of the most basic but vital tools is learning how to interact and communicate effectively [42] in order to promote and encourage more effective communication between the providers and receivers of language support at schools, (1) awareness, (2) understanding and (3) knowledge are key factors. Namely, pre-service and in-service teachers, need to learn intercultural awareness and attain skills in intercultural communication competency.

Participant #39 mentions the stereotype some Japanese teachers have about foreign students at the school and their linguistic identities:

*There are only a few children who I work with whose mother tongue is English. Recently I worked with children whose mother tongues are languages such as Tagalog, Chinese, Vietnamese and Indonesian. When I tell people that I teach Japanese to foreign children, they ask me if I teach them in English. School teachers also have this stereotype that all foreigners can speak and understand English* (Participant #39).

What Participant #39 is describing is the school community's lack of awareness and knowledge about the linguistic diversity of the foreign children at the school. If teachers

are not made aware of the linguistic backgrounds of the EMC, there will be little chance of the school community (fellow students, school staff and management, local community, etc.) being able to have a deeper understanding of both the language and learning needs of the children.

> *When foreign children come to enroll at the school, the first thing teachers think that they needed to do to prepare is download Google Translate application on their phones or that they have learn about the culture of that child. However, I think first of all they need to learn how to use simplified, clear and easy-to-understand Japanese so that it will be easier for the child to understand and communicate. ... There are so many schools that don't understand this and speak too fast, try to speak English to the child (which may not even understand it). It means a lot to the young child to be spoken in Japanese and feel included and cared for, and in turn, affects their motivation to learn and communicate in Japanese.* (Participant #39)

There is a recognized need for schools to at least learn about intercultural awareness in school, learning how to be better aware of cultural, linguistic diversity, interaction and communication skills.

The first participant whom I interviewed for this research project explained it quite simply when asked what they thought to be the most fundamental thing to be done in order to promote better learning support for ethnic minority children.

> *As for support, I think that the easiest and most fundamental thing the country, the city needs to is first of all educate school children and school community that that it is okay to be different, and that everyone does not have to be the same: not all Japanese need to be the same, and there are many different Japanese in Japan. Diversity education is needed to be taught at schools for children, school teachers and parents* (Participant #1).

Memo 3 (21 February 2018)

According to the comments of the representatives of governmental organizations and NGOs I interviewed, information on the topic of intercultural education and/or training on how to teach JSL to students is included in curriculums for pre-service teachers and in-service teachers who are renewing their teaching licenses. However, the actual depth of the content was described as surface level, and limited time was allotted to focus on JSL teacher training and intercultural intelligence skill training.

To what extent knowledge and skills in intercultural education and intercultural communication diversity education will be implemented in teacher training curriculums is yet to be seen. For example, there are opinions among language support teachers that it would be beneficial for homeroom teachers, in particular, to be made to attend the training sessions on providing Japanese language support.

> *We (language support teachers) have about 6 training sessions per year. I always tell the organizers that the training sessions should make it a rule for it be mandatory for homeroom teachers who have no idea what to do when a foreign child suddenly enrolls in the school and is put in their care in the classroom. However, the organizers don't consider making such a rule as it seems difficult to apply.* (Participant #41)

However, according to the discussions with both Board of Education officials and public school authorities, in order to decrease the workload overload of teaching staff and frequency of teacher meetings, the number of workshops and faculty development sessions have been decreased substantially. This across-the-board decision to cut the number and frequency of training opportunities is making it difficult to enforce any type of mandatory training sessions for in-service teachers.

## 4. Conclusions

This paper explores the interconnected themes voiced by language support teachers employed by both the City M Board of Education and public schools, who are actively

engaged in providing essential learning assistance to ethnic minority children. Throughout the exploration, it becomes evident that despite their diligent utilization of resources provided by the Ministry of Education and the Board of Education, significant challenges in the form of constrained budgets, limited teaching resources, time constraints, and imposed restrictions persistently impede their endeavors.

Moreover, the issue extends beyond mere resource limitations, encompassing a notable deficiency in the understanding of the distinctive learning needs of ethnic minority children. The interaction and communication skills of both school teachers and students further compound the complexity of the situation. It becomes increasingly clear that a comprehensive solution requires proactive initiatives from individuals in senior school leadership positions such as heads of schools and school principals.

In light of the findings of this research, it is imperative that senior school leadership assumes a pivotal role in fostering a holistic approach to addressing the challenges at hand. Creating platforms for inter-cultural and social justice awareness and understanding, particularly among in-service educators, emerges as a crucial step forward. Simultaneously, Boards of Education bear the responsibility of instigating a shift in the perspective of school senior leadership teams. By promoting a heightened awareness of their roles in policy review, implementation, and reform, the groundwork can be laid for a more prepared and equipped educational environment. This collective effort ultimately paves the way for the effective integration and education of future ethnic minority students within their academic communities.

*Limitations*

The objective of this research was not to comprehensively cover all the social or educational issues of every category of a child who is considered an ethnic minority in City M. Rather, the intent was to explore the language and learning support provided to ethnic minority children and the agencies who support these children—public elementary schools, government and non-government support agencies, learning support volunteers, and grassroots organizations—by examining people's experiences and perspectives. In conducting this study, I was unable to gain access to public elementary school classrooms to directly observe ethnic minority school children learning together with their learning support teachers. I chose not to interview children themselves, due to ethical issues as well as the challenges of accurately interpreting their feelings and opinions. Although direct interviews with children were not feasible within the scope of this study, the voices of the ethnic minority children were effectively channeled through the research participants I engaged with. These participants, specifically schoolteachers and language support volunteers, emerged as a medium for capturing the experiences, emotions, and perspectives of the children whom they teach. Through our conversations, I was able to clearly perceive the rich tapestry of these children's encounters and sentiments.

The implementation of the culturally responsive sustaining education framework can promote the advancement of language equity by enhancing Japanese language learning support by incorporating diverse cultural backgrounds and ensuring that educational practices resonate with students' individual experiences. By doing so, it creates an environment where Japanese language learning is not only effective but also culturally relevant. Through this approach, students from various linguistic backgrounds can engage more meaningfully with the learning process, leading to improved Japanese language acquisition, increased involvement in their social activities both within and outside of the school environment, and ultimately, fostering greater language equity in local communities in Japan.

Schools need to become proactive in becoming self-sufficient public educational institutions in making priority to educate their staff and students, as well as to provide comprehensive social justice education to their teachers. By doing so, schools can establish a continuous cycle of knowledge-sharing that extends to future generations of both teachers and students. Taking this into account, schools have a fundamental responsibility to allocate resources towards not just enhancing learning support for Japanese as a Second Language,

but also towards robust professional development programs in diversity education for their educators. In this context, the dedication of senior school leadership to providing teachers and staff with specialized knowledge becomes crucial. Only through such a commitment can teachers be fully equipped to provide comprehensive support for EMC, enabling students not only to achieve their linguistic objectives within the school but also to cultivate skills that will propel them to excel in various contexts beyond the classroom.

**Funding:** This research received no external funding.

**Institutional Review Board Statement:** The study was conducted in accordance with the Declaration of Helsinki, and approved by the Research Ethics Committee of Kyosei Studies, Graduate School of Human Sciences, Osaka University (protocol code OUKS1731, approved 31 October 2017)." for studies involving humans.

**Informed Consent Statement:** Informed consent was obtained from all subjects involved in the study.

**Data Availability Statement:** Data sharing is not applicable to this article due to privacy and ethical restrictions.

**Conflicts of Interest:** The author declares no conflict of interest.

## Abbreviations

| | |
|---|---|
| BOE | Board of Education (usually prefectural or city level) |
| CRSE Framework | Culturally Responsive-Sustaining Education Framework |
| EMC | Ethnic Minority Children |
| JSL | Japanese as a Second Language |
| IC | International Classroom (a classroom designated for teaching Japanese as a Second Language to students who need language support) |
| MEXT | Ministry of Education, Culture, Sports, Science and Technology. MEXT was established in 2001; until that time the government ministry of charge of education was the Ministry of Education. |
| MTLS | Mother Tongue Language Supporter |

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
