# Peer review of "Stakeholders’ Experiences and Perceptions of the Provision and Practice of Language Support for Ethnic Minority School Children in Japan"

_societies, doi:10.3390/soc13090197_

Round 1
Reviewer 1 Report
The paper is generally well written and coherent. The study is well designed, with the inclusion of various 'stakeholders' who could potentially contribute towards the successful implementation of language support programs. The comprehensive coverage of the 'eco-system' surrounding such programs allows trusty valuable recommendations to be made. There is a clear explanation of the methodology from pre data collection to the analysis stage and also consideration of the positionality of the researcher in a qualitative study.
The discussion of findings is sound and backed up by strong data evidence from the participants. The study provides good data for the improvement of language support system for ethnic minorities in Japan and that should be the most important significance of the study. But I wonder if you can also situate the study in wider literature and thus draw further implications from the study. If so, the significance of the study could be extended.
For future research, perhaps there could be a focus on the students' voices and perspectives. The author shared an observation about signages made by the students themselves. Giving them the space to express themselves also give educators and volunteers a chance to get to know their world a little more.
Do take note of the following:
- Alignment of tables within the text and font size within the tables
- Do a quick check on punctuations, spacing and missing words (e.g. 'a' or 'the') The following points are just some I caught.
- page 10 - two full-stops on line 390.
page 20 - line 790 'a' system of Japanese learning support
- line 808 double full-stops.
page 21 - using 'they' to refer to Participant 1
Author Response
Dear Reviewer,
Thank you very much for your careful review of and helpful feedback on my paper.
Below are my responses.
- Alignment of tables within the text and font size within the tables      →I have corrected the allignment to the best of my ability, but due to the differences in allignment settings and default settings on both the author's and the reviewer's computers and the online submission platform, some of the allignment of the tables and text has been altered and changed beyond my control. I do hope that the final editors for the journal will be able to correct and remedy this. Thank you very much for your assistance in advance.
- Do a quick check on punctuations, spacing and missing words (e.g. 'a' or 'the') The following points are just some I caught.            → I have remedied this to the best of my ability.
- page 10 - two full-stops on line 390.→corrected.
page 20 - line 790 'a' system of Japanese learning support→corrected.
- line 808 double full-stops.→corrected.
page 21 - using 'they' to refer to Participant 1 →I have chosen the pronoun "they" to refer to Participant #1 and all other participants in this research study in order not disclose their genders and therefore to protect the identity of the participants in my study.
Reviewer 2 Report
The article entitled "Stakeholders' experiences and perceptions of the provision and practice of language support for ethnic minority school children in Japan" fits into the monograph "Migration and Multilingual Education: An Intercultural Perspective" as it addresses part of the topics proposed in the monograph. Focusing on a specific geographical area, it explores perceptions and experiences of language support for minorities in schools. An attempt is made to study language learning situations in these institutions; and, likewise, the opinion of significant agents who have an impact on the education of these minorities is explored in depth.
The methodology is based on a descriptive study using interviews with nine participants. The analysis with Nvivo is correct and all the answers from this sample are well analysed. The conclusions are coherent and clearly written.
However, the text is too focused on the Japanese environment and would require a number of modifications for publication:
a) Extend the references used
a. There are only 23 references and they are very focused on the context of the study.
b. More international references on language teaching; minorities; the importance of language; language and cultural policy, etc., should be cited.
c. Currently only 5/23 references are from the last five years. At least 25% of the final total of references (including new ones) should be from that period (2018-2023).
b) If the perspective of the monograph is based on interculturality and diversity, the text should include, both in its theoretical justification and in its conclusion, allusions to the international context in a comparative manner. This would enable the article to have a greater impact and dissemination.
c) The objectives of the study should be made more explicit.
d) A section on the discussion of results should be included where the degree of achievement of objectives and the impact of theoretical authors can be indicated in contrast to the results achieved in this research.
e) The conclusions section could be expanded to include a reflection on the limitations that have already been made at the beginning of the article (it is possible that the most advisable place for them is in this conclusions section).
Author Response
Dear Reviewer,
Thank you very much for your review of my paper. I genuinely appreciate all of your thoughtful suggestions and the opportunity to consider alternative perspectives.
My responses are below:
However, the text is too focused on the Japanese environment and would require a number of modifications for publication.→ The sole purpose of this paper is to focus deeply and exclusively on the issues of Japanese language support in Japan.
a) Extend the references used→ I have tried my utmost best to research and reference a wide variety of references which were the latest literature on my subject at the time when was researching and writing this paper.
a. There are only 23 references and they are very focused on the context of the study→ I have used references not only focusing on the context of the study but on other issues and subjects related to the context of the study, which I feel is very important in order to understand the related issues of my study.
b. More international references on language teaching; minorities; the importance of language; language and cultural policy, etc., should be cited.→I believe the the case in Japan is quite unique and cannot be simply compared nor closely referred to other examples of cultural policy or general language teaching of other countries.
c. Currently only 5/23 references are from the last five years. At least 25% of the final total of references (including new ones) should be from that period (2018-2023).→ I conducted all of my researched and and wrote this paper during 2018-2020 and feel that the references used at the time of writing this paper are sufficient.
b) If the perspective of the monograph is based on interculturality and diversity, the text should include, both in its theoretical justification and in its conclusion, allusions to the international context in a comparative manner. This would enable the article to have a greater impact and dissemination.→I greatly appreciate the reviewer's comment and acknowledge your opinion. However,the intention of the monograph was to provide a deep exploration of a particular cultural context rather than a comparative analysis. I feel that if I explored multiple international contexts in a comparative manner might dilute the depth of analysis and limit the ability to present a comprehensive understanding of the specific issue of Japanese language support practices.
c) The objectives of the study should be made more explicit.→I personally feel as the author of this paper that the objectives of the study were made quite explicit to the reader in the beginning and throughout this paper.
d) A section on the discussion of results should be included where the degree of achievement of objectives and the impact of theoretical authors can be indicated in contrast to the results achieved in this research.→Thank you for your feedback and suggestions regarding the inclusion of a section on the discussion of results. While I appreciate your perspective, I respectfully disagree with the need for a direct comparison of the degree of achievement of objectives and the impact of theoretical authors in contrast to the results of this research. While a section on the discussion of results is indeed valuable in many studies, in this particular monograph, I believe that such a comparison could potentially deviate from the main focus and dilute the depth of analysis within the chosen context.
e) The conclusions section could be expanded to include a reflection on the limitations that have already been made at the beginning of the article (it is possible that the most advisable place for them is in this conclusions section).
→Thank you for your feedback regarding the expansion of the conclusion section to include a reflection on the limitations. In this paper, the limitations were intentionally addressed at the beginning to provide transparency and clarity to the readers regarding the scope and boundaries of the research. I feel that placing them in the conclusions section might risk confusing the readers or undermining the initial framework established in the introduction.
If you have any further recommendations or if there are specific aspects you believe should be addressed, I would be open to discussing them further. Thank you again for your valuable feedback.
Round 2
Reviewer 2 Report
A re-examination of the article shows that the changes suggested by the evaluator have not been implemented. Each of the proposals is always responded to with arguments that maintain that everything is fine. From the evaluator's point of view, aspects such as the references (updating them), the internationalisation of the article, the length of the conclusions, or the narrowing of the focus, are key to improvement and are not specific issues, but are in line with the guidelines of the Scientific Community on the preparation of research articles. The answers given by the author of the article disagree with many of the issues expressed by the evaluator. While this position is respectable, it is also respectable that the reviewer maintains his assessment.
Author Response
Dear Reviewer,
Thank you very much for your re-examination of the manuscript. I greatly appreciate the time and care that the reviewer has invested in reviewing this article.
I have responded to the best of my knowledge and ability to the Reviewer's comments:
- (Reviewer's comments:) There are only 23 references and they are very focused on the context of the study / b. More international references on language teaching; minorities; the importance of language; language and cultural policy, etc., should be cited / c. Currently, only 5/23 references are from the last five years. At least 25% of the final total of references (including new ones) should be from that period (2018-2023) / More international references on language teaching; minorities; the importance of language; language and cultural policy, etc., should be cited. (Author's response:)As suggested by the Reviewer, I have added 14 new international references and several new citations published between the period of 2018-2023 on language teaching, minorities and language and cultural policy, which are highlighted in yellow.
- (Reviewer's comment: )If the perspective of the monograph is based on interculturality and diversity, the text should include, both in its theoretical justification and in its conclusion, allusions to the international context in a comparative manner. This would enable the article to have a greater impact and dissemination. (Author's response:)As suggested by the Reviewer, I have made sure to include a the applied theoretical framework to this study, both in a separate section explaining the applied theoretical framework, as well as including a justification in the conclusion section, which is highlighted in yellow throughout the paper.
- (Reviewer's comment:)The objectives of the study should be made more explicit. (Author's response:) As suggested by the Reviewer, I have attempted to make the objectives of this study more explicit by both revising and expanding my explanation in the Purpose of Study section of this manuscript, which is highlighted in yellow.
- (Reviewer's comments:) d.A section on the discussion of results should be included where the degree of achievement of objectives and the impact of theoretical authors can be indicated in contrast to the results achieved in this research. / e.The conclusions section could be expanded to include a reflection on the limitations that have already been made at the beginning of the article (it is possible that the most advisable place for them is in this conclusions section). (Author's response:) As suggested by the Reviewer, I have revised the Conclusion section where the impact of theoretical framework is indicated. I have also relocated the Limitations section as part of the Conclusion section. I have also included a reflection in the Limitations section, as suggested by the Reviewer.
_______________
Thank you very much again for your valuable feedback and for offering your expertise in the review of my manuscript.
Respectfully yours,
The Author
Round 3
Reviewer 2 Report
The article has been amended in different sections. The result of these rectifications is satisfactory. Therefore, from the evaluator's point of view, the text can be accepted.